# Longitudinal Analysis of 1α,25-dihidroxyvitamin D_3_ and Homocysteine Changes in Colorectal Cancer

**DOI:** 10.3390/cancers14030658

**Published:** 2022-01-28

**Authors:** Dorottya Mühl, Magdolna Herold, Zoltan Herold, Lilla Hornyák, Attila Marcell Szasz, Magdolna Dank

**Affiliations:** 1Division of Oncology, Department of Internal Medicine and Oncology, Semmelweis University, 1083 Budapest, Hungary; muhl.dorottya@med.semmelweis-univ.hu (D.M.); herold.zoltan@med.semmelweis-univ.hu (Z.H.); hornyak.lilla@med.semmelweis-univ.hu (L.H.); szasz.attila_marcell@med.semmelweis-univ.hu (A.M.S.); 2Department of Internal Medicine and Hematology, Semmelweis University, 1088 Budapest, Hungary; herold.magdolna@med.semmelweis-univ.hu

**Keywords:** vitamin D, homocysteine, colorectal neoplasms, longitudinal analysis, survival analysis

## Abstract

**Simple Summary:**

Vitamin D_3_ and homocysteine level abnormalities are both strongly related to colorectal cancer (CRC) etiology. The aim of this retrospective study was to investigate the longitudinal change in these two parameters and the relationships between the two, in addition with other clinicopathological and laboratory parameters. A swoosh-shaped trend was observed for the change in serum homocysteine levels of all of the CRC patients. The circulating vitamin D_3_ level was constant or increased in those patients without metastasis. After an initial increase, the disease-worsening effect of metastases cancelled out all of the positive effects of vitamin D_3_ in metastatic patients, even despite its continuous supplementation. Right-sided tumors, male sex, and the pathological values of serum lipids, albumin, total protein, and inflammatory markers were associated with lower vitamin D_3_ and higher homocysteine level. Based on our results, we recommend a modified vitamin D_3_ supplementation regimen for metastatic CRC, which includes laboratory measurement-based titration.

**Abstract:**

Background: 1α,25-dihydroxycholecalciferol (1,25(OH)_2_D_3_) and homocysteine are known to play a role in the pathophysiology of colorectal cancer (CRC). In health, the two changes are inversely proportional to each other, but little is known about their combined effect in CRC. Methods: The serum 1,25(OH)_2_D_3_ and the homocysteine levels of eighty-six CRC patients were measured, who were enrolled into four cohorts based on the presence of metastases (Adj vs. Met) and vitamin D_3_ supplementation (ND vs. D). Results: 1,25(OH)_2_D_3_ was constant (Adj-ND), increased significantly (Adj-D, *p* = 0.0261), decreased (Met-ND), or returned close to the baseline after an initial increase (Met-D). The longitudinal increase in 1,25(OH)_2_D_3_ (HR: 0.9130, *p* = 0.0111) positively affected the overall survival in non-metastatic CRC, however, this effect was cancelled out in those with metastasis (*p* = 0.0107). The increase in homocysteine negatively affected both the overall (HR: 1.0940, *p* = 0.0067) and the progression-free survival (HR: 1.0845, *p* = 0.0073). Lower 1,25(OH)_2_D_3_ and/or higher homocysteine level was characteristic for patients with higher serum lipids, albumin, total protein, white blood cell and platelet count, male sex, and right-sided tumors. No statistically justifiable connection was found between the target variables. Conclusions: A measurement-based titration of vitamin D_3_ supplementation and better management of comorbidities are recommended for CRC.

## 1. Introduction

Colorectal cancer (CRC) is the third most commonly diagnosed cancer type with almost 1.9 million new CRC cases being discovered worldwide annually, and CRC is responsible for 916,000 deaths, being the second leading cause of cancer deaths, according to the GLOBOCAN 2020 data [1]. Low circulating 1α,25-dihydroxycholecalciferol (synonyms are as follows: calcitriol, dihydroxy vitamin D_3_, and vitamin 1,25(OH)_2_D_3_; the biologically active form of vitamin D_3_) and high homocysteine levels are characteristic for CRC [2,3,4]. In the last decades, a large number of observational and randomized clinical trials have investigated the effect of vitamin D on CRC and associations have been proposed between the circulating levels of vitamin D metabolites, CRC incidence, and patient survival [5,6,7,8]. Furthermore, in vitro studies have suggested various anticancer actions of vitamin D, including, but not limited to, anti-proliferative, pro-differentiation due to gene regulatory changes, pro-apoptotic, angiogenesis inhibitory, immune modulatory, and regulatory effects on microRNAs and immune cells [5,9,10,11]. It has been previously reported that, after primary tumor removal, the high prevalence of vitamin deficiency among CRC patients somewhat normalizes and the largest and fastest increase have been observed in those who are on adjuvant vitamin D supplementation, as expected [12,13,14].

In contrast to vitamin D, the relationship between homocysteine and CRC is a less investigated area. Homocysteine is an essential sulfur-containing amino acid produced by various cells throughout the human body by demethylation of methionine. In healthy individuals, its circulating levels are low (5–15 µmol/L, which is somewhat lower in women) [15,16] and inversely proportional to the amount of hydroxyvitamin D_3_ in the blood [16]. Higher homocysteine level had been found to be a risk factor for the increased incidence of CRC [17,18,19] and within the first year after primary tumor removal surgery, Ni et al. [20] reported a slow but constant decreasing level of homocysteine in CRC patients without tumor recurrence, while an increase has been found in those with relapse.

Despite the large and increasing number of studies investigating vitamin D_3_ or homocysteine in CRC, to our knowledge, no study investigated the single-time nor the longitudinal relationships between serum 1,25(OH)_2_D_3_ and homocysteine levels in CRC, nor if their changes are affected by any other laboratory or clinicopathological parameters. Therefore, a retrospective longitudinal observational cohort study was created to investigate the combined effect of vitamin 1,25(OH)_2_D_3_ and homocysteine on CRC. The possible associations between the two parameters and the various clinicopathological properties, comorbidities, other laboratory measurements, and the anamnestic data were further objectives of the investigation.

## 2. Materials and Methods

The study was conducted in accordance with the World Medical Association’s Declaration of Helsinki; handling of patient data was in accordance with the General Data Protection Regulation issued by the European Union. The study was approved by the Regional and Institutional Committee of Science and Research Ethics, Semmelweis University (SE TUKEB 133/2015).

### 2.1. Study Design and Patient Selection

A retrospective longitudinal observational cohort study was conducted with the inclusion of eighty-six CRC patients who attended at the Division of Oncology, Department of Internal and Medicine and Oncology, Semmelweis University, Budapest, between 2017 and 2018. Diagnosis of CRC was given by histopathological examination of colonoscopic biopsy or surgical specimens. Inclusion criteria of the study required at least one vitamin 1,25(OH)_2_D_3_ and homocysteine measurement, performed at the Central Laboratory of Semmelweis University. Exclusion criteria included the following: age < 18 years, histopathological diagnosis of other tumor forms than adenocarcinoma, any previous malignancies, known hematologic, inflammatory bowel, systemic autoimmune, mental and/or inadequately controlled thyroid diseases, the usage of systemic corticosteroids 90 days prior to the baseline visit date, erythropoiesis-stimulating agents, and patients with an Eastern Cooperative Oncology Group (ECOG) performance status >2.

Of the eighty-six patients, a total of 433 visits (5 visits/patient on average) had been processed. The first visit was performed after the primary tumor removal surgery, if technically possible in those of with metastasis, and prior the first adjuvant or metastatic chemotherapy administration in those patients without and with metastasis, respectively. Patients attended at the second and third visits one and two months after the baseline measurement, respectively. Later measurements were performed within 2–3 month intervals.

### 2.2. Clinicopathological and Laboratory Data Measurements

Anamnestic data, including recent medications and comorbidities, were collected at the time of the baseline measurements, while laboratory measurements from fasting blood samples were recorded during every visit. Complete blood count, aspartate and alanine transaminase, gamma-glutamyl transferase, lactate dehydrogenase, creatinine, high-sensitivity C-reactive protein, cholesterols and triglycerides, and tumor markers were measured at the Central Laboratory of Semmelweis University. Estimated glomerular filtration rate (eGFR) was calculated using the Chronic Kidney Disease-Epidemiology Collaboration equations [21]. Serum vitamin 1,25(OH)_2_D_3_ and homocysteine was measured using the DiaSorin LIAISON^®^ XL 1,25-dihydroxyvitamin D chemiluminescent immunoassay (DiaSorin S.p.A., Saluggia, Italy) and the Roche Cobas^®^ HCYS Homocysteine Enzymatic assay (Roche Diagnostics GmbH, Mannheim, Germany) kits, respectively. To avoid degradation, vitamin D samples were taken and stored in light-protected collection tubes until measurement.

CRC staging was given by histopathological examination of surgical specimens and imaging studies; the American Joint Committee on Cancer grouping was used [22]. Side of CRC was described as right-sided if the tumor was originating from cecum, ascending colon, and proximal two-third of the transverse colon; and left-sided if originating from the distal one-third of the transverse colon, descending colon, sigmoid colon, and rectum [23]. Chemotherapy was grouped as adjuvant if no metastasis and as first-line, second-line etc. if metastasis was present Selection of chemotherapy protocol(s) was based on national and international guidelines. Vitamin D_3_ supplementation was prescribed for patients on platinum-based chemotherapies and supplementation was also recommended for the remaining patients. Vitamin D_3_ supplementation (3000 IU/day) was recorded based on digital prescription records and paper-based patient records. Overall (OS) and progression free (PFS) survival of patients was defined as the length of time from the date of study inclusion until the death and until disease progression or death, respectively. The cause of death was recorded as two separate events if patients died due to CRC-related or non-cancer related reasons. Follow-up of patients was terminated on 30 September 2021 and patients alive at this time were right-censored. The RECIST guideline v1.1 was used to evaluate response to treatment [24].

### 2.3. Statistical Analysis

Statistical analysis was performed with R version 4.1.1 (R Core Team, 2021, Vienna, Austria). Kruskal–Wallis test with *p*-value corrected pairwise Wilcoxon–Mann–Whitney U-tests as post hoc, Fisher’s exact test and Spearman rank correlation was used for comparisons between cohorts. To detect the changes of various parameters in time, natural cubic spline adjusted random intercept linear mixed-effect models were used (R-package “nlme”, developed by Pinheiro, Bates DebRoy, Sarkar and the R Core Team, version 3.1-153). Survival models were calculated for longitudinal data using competing risk Cox regression models with time-dependent covariate(s) (R package “survival”, developed by Therneau and Grambsch, version 3.2-13). *p* < 0.05 was considered as statistically significant and *p*-values were corrected with the Holm method [25] for multiple comparisons problem. Results were expressed as mean ± standard deviation, the number of observations (percentage), and as hazard ratio (HR) with 95% confidence interval (95% CI) for continuous, count, and survival data, respectively. Naïve Kaplan–Meier survival curves were drawn with the R-package “survminer” (developed by Kassambara, Kosinski and Biecek, version 0.4.9).

## 3. Results

A total of 86 CRC patients were included in the study. The patients were divided into four cohorts based on the presence of metastases and whether they were on vitamin D_3_ supplementation during our observation time or not. Nineteen, twenty-five, ten, and thirty-two patients were assigned to the ‘adjuvant without vitamin D_3_ supplementation’ (Adj-ND), ‘adjuvant with vitamin D_3_ supplementation’ (Adj-D), ‘metastatic without vitamin D_3_ supplementation’ (Met-ND), and to the ‘metastatic with vitamin D_3_ supplementation’ (Met-D) groups, respectively. The anamnestic data of study subjects are summarized in Table 1.

### 3.1. Baseline Patient Characteristics and Measurements

The comparisons between the groups were performed in order to test whether the later-treated and untreated groups differed and for any confounding parameters. Only marginal differences were found both in vitamin 1,25(OH)_2_D_3_ (Kruskal–Wallis: *p* = 0.0749, Figure 1A) and in homocysteine (Kruskal–Wallis: *p* = 0.0535, Figure 1B) levels. It was tested whether other factors, such as sex, staging, location of the tumor, and the various comorbidities, such as type 2 diabetes or hypertension, affected the parameters outlined above. Vitamin 1,25(OH)_2_D_3_ was not affected by any of these grouping factors, while homocysteine was significantly higher if the tumor was located on the right side of the colon (left-sided: 12.48 ± 3.94 ng/mL, right-sided: 15.34 ± 5.65, *p* = 0.0174).

The comparison of laboratory results between the four cohorts revealed only the differences related to the presence of metastasis. The pathological values of high-sensitivity C-reactive protein (Kruskal–Wallis test, *p* = 0.0025), lactate dehydrogenase (*p* = 0.0004), carcinoembryonic antigen (*p* < 0.0001), and carbohydrate antigen 19-9 (*p* = 0.0080) were found in the study subjects with metastasis, compared to those CRC patients without metastasis (Table 2). Prior to any vitamin D supplementation, no differences could have been justified between the groups Adj-ND vs. Adj-D and Met-ND vs. Met-D.

The correlation analysis of laboratory results of vitamin 1,25(OH)_2_D_3_ and homocysteine revealed only previously found relationships [26,27,28,29,30,31], such as higher vitamin 1,25(OH)_2_D_3_ levels, which are associated with lower but still within the normal range platelet counts (Spearman’s rho: −0.25, *p* = 0.0213), total cholesterol (Spearman’s rho: −0.25, *p* = 0.0182), and low-density lipoprotein levels (Spearman’s rho: −0.27, *p* = 0.0130). Higher homocysteine levels were associated with lower eGFR values (Spearman’s rho: −0.38, *p* = 0.0003) and higher serum creatinine levels (Spearman’s rho: +0.34, *p* = 0.0013). A marginal association was found between homocysteine and white blood cell counts (Spearman’s rho: +0.21, *p* = 0.0533). Furthermore, significantly lower homocysteine level could have been observed with higher red blood cell counts (Spearman’s rho: −0.35, *p* = 0.0223) within the patients without metastasis, compared to those with metastasis (*p* = 0.3311).

### 3.2. Analysis of Longitudinal Data

In order to determine whether the vitamin 1,25(OH)_2_D_3_ and homocysteine levels changed differently in the four cohorts, with respect to the course of CRC, we chose the following approach. Natural cubic spline adjusted random intercept linear mixed-effect models were constructed, where all of the baseline and further repeated measurements from all of the 86 study participants were used. A total of 417 and 421 vitamin 1,25(OH)_2_D_3_ and homocysteine measurements were collected, respectively. The model prediction intervals where an insufficient number of observations were available, were omitted as the accuracy of these model estimates would be insufficient with missing or a low number of samples. It was found that the vitamin 1,25(OH)_2_D_3_ level of the Adj-ND patients fluctuated around an approximately constant value (*p* = 0.2640), while a constant increase was observed in those patients of the Adj-D cohort (*p* = 0.0261). A constant decrease was found for the Met-ND, while a subdued increase was found for the Met-D cohorts respectively (Figure 2A). In the case of homocysteine, a swoosh shape (sharp decline with slow recovery) was predicted for all of the study groups. Except for the deviation within the baseline value of the Met-D cohort (*p* = 0.0195), the only other difference between the cohorts was that the second half of the group averages resembled a saturation curve in those with vitamin D_3_ supplementation, while a steady increase in homocysteine level was observed in the non-supplemented patients (Figure 2B). It must be noted, however, that although these pattern differences were assumed to be clinically different, the curves were not statistically different (*p* = 0.3003).

The effect of the various clinicopathological and laboratory parameters on the changes in the vitamin 1,25(OH)_2_D_3_ and homocysteine levels were also investigated. The effect of staging was basically the same as belonging to any of the with or without metastasis groups (vitamin 1,25(OH)_2_D_3_: *p* = 0.3363, homocysteine: *p* = 0.6276, Appendix A). Right-sided tumors were associated with constantly higher homocysteine (*p* = 0.0041) but with similar vitamin 1,25(OH)_2_D_3_ (*p* = 0.6777) levels (Figure 3). Those patients within the Adj-ND and Adj-D groups, in whom metastasis developed at a later stage, after an initial similarity, tendentiously lower vitamin 1,25(OH)_2_D_3_ (*p* = 0.1030, Figure 4A) levels occurred, compared to those who were metastasis-free until the end of our observations. Similarly, the vitamin 1,25(OH)_2_D_3_ levels decreased after an initial similarity in those patients who had progressive disease (*p* = 0.0731, Appendix A) or who had died (*p* = 0.0267, Figure 5A). In contrast, the homocysteine levels were not affected statistically, neither by later metastasis (*p* = 0.6813, Figure 4B), disease progression (*p* = 0.5225, Appendix A), nor by death (*p* = 0.2153, Figure 5B). Hypertension (*p* = 0.0585, Appendix A) and other cardiovascular diseases (*p* = 0.0672, Appendix A) caused only tendentiously higher vitamin 1,25(OH)_2_D_3_ levels, while any previous cardiovascular event(s), type 2 diabetes mellitus, or thyroid diseases (in an euthyroid state) did not affect the change in the two parameters (Appendix A). The sex of patients marginally affected the change in vitamin 1,25(OH)_2_D_3_ levels (*p* = 0.0878) and a marginally constant difference between the homocysteine levels of the two sexes was observed throughout the study (*p* = 0.0803). For both cases, the clinically advantageous effect was observable in women (Figure 6).

Despite previous literature data on a healthy population [16] and their seemingly proportionate change in opposite directions in the first months of daily 3000 IU vitamin D_3_ supplementation (Figure 2), no effect of vitamin 1,25(OH)_2_D_3_ over homocysteine (*p* = 0.5197), and vice versa (*p* = 0.6625), could have been justified, even if only the first periods of our observation (vitamin 1,25(OH)_2_D_3_: *p* = 0.1301, homocysteine: *p* = 0.2641) were studied. By examining the longitudinal relationships between the study and laboratory parameters, we could define the following two different types of trends. In the first, the changes in the laboratory parameters over time did not affect the changes in the 1,25(OH)_2_D_3_ and homocysteine levels, but a constant difference related to pathological values could have been observed throughout the study. In contrast, in the other case, where an interaction was found between the two variables, the change in one of the two parameters had a significant effect on the change in the other. The first type of the two trends was observed for the serum total cholesterol (*p* < 0.0001), low density-lipoprotein cholesterol (*p* < 0.0001), and for triglycerides (*p* < 0.0001); in all cases, constant high levels were associated with lower 1,25(OH)_2_D_3_ levels. Higher homocysteine levels were associated with constantly higher white blood cell counts (*p* = 0.0092), platelet counts (*p* = 0.0092), serum aspartate aminotransferase (*p* = 0.0021), lactate dehydrogenase (*p* = 0.0009), total cholesterol (*p* = 0.0146), low-density lipoprotein cholesterol (*p* = 0.0441), and creatinine (*p* < 0.0001) levels, and with constantly lower eGFR rates (*p* < 0.0001). Interaction was found between the change in 1,25(OH)_2_D_3_ levels and the change in serum albumin (*p* = 0.0158). Over time, as albumin levels normalized, the speed of vitamin 1,25(OH)_2_D_3_ saturation slowed down. The change in homocysteine was significantly affected by the changes in the serum total protein (*p* = 0.0072). Normalized levels of serum total protein were more likely to be associated with constant homocysteine levels at the end of the observation period. In addition to investigating the univariate effects of the laboratory parameters, their combined multivariate effect was also analyzed, where the first type of influencing effects made the interaction terms statistically not significant. No effect could have been justified for the remaining laboratory parameters, which are listed in Table 2.

### 3.3. Survival Analysis

The overall and progression-free survival of patients was investigated. In the former, the following two endpoint events had been defined: (1) death related to CRC and (2) death related to natural causes (e.g., old age) or other, non-cancerous diseases. The first event occurred in 36 (41.9%) and the second in 4 (4.7%) cases, while the follow-up of the survivors was terminated on 30 September 2021. Two and three patients in the non-metastatic groups died due to CRC-related or other causes, while 34 and 1 within the metastatic groups died, respectively. Disease progression occurred in 3, 4, 9, and 28 cases of the Adj-ND, Adj-D, Met-ND, and Met-D cohorts, respectively. Non-adjusted, naïve overall, and progression-free Kaplan–Meier survival curves of the four study cohorts are shown in Figure 7.

The effect of longitudinal vitamin 1,25(OH)_2_D_3_ and homocysteine changes in disease-specific overall and progression-free survival was analyzed via competing risk Cox regression models. Similar to the results of the mixed effect models described in Section 3.2, different HRs were assumed for the four cohorts due to the different trends in the change the in vitamin 1,25(OH)_2_D_3_ levels. No adjustment was required for homocysteine, as its change was similar in all cohorts.

Being in any of the metastatic groups, as expected, significantly worsened patients’ life expectancy. Increasing levels of serum vitamin 1,25(OH)_2_D_3_ had a positive effect on the overall survival of the patients without metastasis (HR: 0.9130, 95%CI: 0.8511–0.9794, *p* = 0.0111), however, the survival-reducing effect of metastases practically cancelled out this positive effect (*p* = 0.0107, adjusted HR of vitamin 1,25(OH)_2_D_3_ of the Met-ND group: 1.0034, 95%CI: 0.8698–1.1567), even with daily 3000 IU vitamin D_3_ supplementation (*p* = 0.0224, adjusted HR of vitamin 1,25(OH)_2_D_3_ of the Met-D group: 0.9979, 95%CI: 0.8622–1.1567). No association was found between disease progression and vitamin 1,25(OH)_2_D_3_ changes (HR: 0.9868, 95%CI: 0.9482–1.0270, *p* = 0.5150), while the increase in homocysteine level was a negative effector of both the overall (HR: 1.0940, 95%CI: 1.0250–1.1680, *p* = 0.0067) and the progression-free survival (HR: 1.0845, 95%CI: 1.0221–1.1510, *p* = 0.0073).

Similar to the independent effects, if both of the parameters were included in an extended model, the increase in serum vitamin 1,25(OH)_2_D_3_ (HR: 0.9173, 95%CI: 0.8567–0.9822, *p* = 0.0133) and homocysteine (HR: 1.0890, 95%CI: 1.0140–1.1690, *p* = 0.0193) level had the same positive and negative effect on the overall survival, respectively. As above, the positive effect of vitamin 1,25(OH)_2_D_3_ was cancelled out by the presence of metastases (Met-ND *p* = 0.0124, Met-D *p* = 0.0420). However, the progression-free survival was only significantly affected by the presence of metastases (Met-ND: *p* = 0.0006, Met-D: *p* = 0.0162), while neither homocysteine (*p* = 0.7757) nor 1,25(OH)_2_D_3_ (*p* = 0.5707) had any effect on disease progression if both of the parameters were included in the same model.

## 4. Discussion

Our knowledge of vitamin D has changed significantly in the last two decades. The latest research has highlighted that vitamin D is acting more similar to a hormone rather than a “simple” vitamin [32]. Vitamin D deficiency develops in most chronic diseases, including various forms of cancers [33]. Although vitamin D supplementation seems to have beneficial effects in various diseases and several observational studies proposed association between cancer mortality and low circulating vitamin D levels, due to the lack of large-scale clinical trials on vitamin D supplementation, insufficient evidence exists on this presumed positive effect of vitamin D over patient survival [8,34]. It has been suggested that supplementation may have a dose-dependent effect on reducing mortality/disease progression [35]. A randomized clinical trial reported marginally better progression-free survival if metastatic CRC patients were treated with higher levels of the active ingredient [36]. In the current study the positive effect of vitamin D supplementation on circulating vitamin 1,25(OH)_2_D_3_ level was basically detectable only in those patients without any metastasis, a sustained increase, and the positive effect of increasing 1,25(OH)_2_D_3_ levels on patient survival was confirmed. In contrast, despite an initial increase, a similar lasting effect could not be achieved in metastatic patients and circulating levels returned to almost the same level as the baseline, simultaneously cancelling out the positive effect on patient survival. It should be mentioned that, based on the available documentation, this trend has been accompanied by a continuous supplementation of daily 3000 IU vitamin D_3_. Taking into account the results of previous studies [35,36] and in light of our results in those patients with metastatic CRC detailed above, we hypothesize that vitamin D supplementation should be regularly supervised with continuous monitoring through the laboratory measurement of vitamin 1,25(OH)_2_D_3_ until a sufficient saturation response is achieved e.g., dosage titration of daily vitamin D should be increased until the serum levels are constantly within the middle of normal range. Our hypothesis is that the tumor burden, accompanied by metastatic CRC, at first is counterweighted via the introduction of vitamin D supplementation, but afterwards the negative effect of the advanced disease slowly completely suppresses the positive effects of vitamin D. The setting of the current study was insufficient to answer this hypothesis, therefore, a prospective study is suggested to properly investigate this effect of metastases over vitamin D supplementation.

Lately, with the possibility of analyzing longitudinal data much more easily, the knowledge about serum level changes in various biomarkers has significantly increased, which includes vitamin D_3_ metabolites as well. In CRC, increasing levels of circulating vitamin D_3_ metabolites have been found in patients whose primary tumor was surgically removed [13], which was accompanied by the decrease in serum high-sensitivity C-reactive protein levels, but no relationship between the two parameters could be justified via mixed effect modeling, similarly to our results. Another study [12] reported increasing hydroxyvitamin D_3_ levels of non-metastatic CRC patients, which, similarly to the results of [13], was preceded by a brief decrease right after tumor resection. The study also reported [12] differences between the hydroxyvitamin D_3_ levels of males and females, patients with and without vitamin D supplementation, and between those patients who received chemotherapy and those who did not Furthermore, increasing serum levels were also associated with better quality of life questionnaire results. The difference between the sexes could have been justified in our study as well. In addition to all of the above, patients, who eventually died as a result of their tumorous disease, had progressive disease, or those who developed a metastasis at a later stage of the disease showed a similar trend in the first interval of our observation to that of survivors, while a decrease was more pronounced in later times. Right-sided tumors were associated with only tendentiously lower levels of serum vitamin 1,25(OH)_2_D_3_ and no difference was found in the lower disease stages (stage II vs. III).

Investigating the effect of various comorbidities on the change in vitamin 1,25(OH)_2_D_3_ was an additional objective of our study. Interestingly, a higher increase in serum levels of vitamin 1,25(OH)_2_D_3_ was more likely to occur in those patients with comorbidities. This observation is presumably due to the fact that comorbid patients are usually under continuous medical supervision with well-established disease treatment prior to CRC as well, which, apart from known disease-related deviations, can also cause their general better condition, compared to those patients without any comorbidities. CRC is known to have a strong relationship with various comorbidities, such as hypertension [37], other cardiovascular diseases including major events [38], and type 2 diabetes mellitus [39]. The observation that comorbid patients are seemingly more protected suggests that the assessment of comorbidities at the time of tumor diagnosis and their appropriate subsequent follow-up may have a positive effect on the outcome of CRC, with high probability. Therefore, the early detection of new, or the balancing of existing comorbidities, in cancer patients is highly recommended, in addition to the routine oncology care.

The relationship between vitamin D_3_ metabolites and cholesterols is controversial. While one study [40] has reported a strong positive correlation between 25-hydroxyvitamin D_3_ and high-density lipoprotein cholesterol, another study [41] has found a significant positive correlation between 25-hydroxyvitamin D_3_ and triglycerides, and total and low-density lipoprotein cholesterol only. In a recent publication [42], a strong connection between various vitamin D_3_ metabolites and serum lipid parameters in CRC has also been justified, suggesting that lower vitamin D levels are more likely to be associated with high lipids. Our results extend these findings, suggesting that low vitamin D levels are inherent in high lipid levels. Another observation from our longitudinal analysis was that there was a very strong connection between albumin and vitamin 1,25(OH)_2_D_3_ levels, which is known for renal diseases [43,44], and in the case of CRC, with the most probability it is connected to disease progression [45].

Another major focus of our study was to further elucidate the role of homocysteine in CRC. Homocysteine related molecular pathways are suggested to have a significant role in the development of CRC [17] and its higher serum level has been found as a risk factor for increased incidence of CRC [17,18,19] and diseases progression [46]. Moreover, a study [20] investigated its change with the course of the disease and it has been found that the homocysteine level reduced from baseline levels in CRC patients without tumor recurrence, while homocysteine levels returned or even exceeded the baseline levels in those patients with tumor recurrence. In contrast to the above, our results revealed a swoosh-shaped trend in the change in the homocysteine level of CRC patients, regardless of any clinicopathological features. The different trends in the change in homocysteine levels between our results and those from the study of Ni et al. are possibly related to the distinct patient selection, as follows: (1) Hungarian patients with European ancestry vs. Chinese patients with Asian ancestry, (2) less rectal cancer cases were included in the present study (25.6% vs. 57.6%), (3) patients younger than 50 years of age were less pronounced (10.5% vs. 29.7%), (4) half of our study subjects were composed of metastatic disease (48.8% vs. less than 40.7%, patients with Dukes’ C or D were not assigned to different cohorts), and (5) the observation period lasted longer in our study (25 vs. 12 months). In addition, higher homocysteine levels were poor prognostic signs for both the overall and the progression-free patient survival.

Analysis of the effect of clinicopathological and laboratory parameters on the change in homocysteine level revealed the previously known relationships that hyperhomocysteinemia is associated with elevated serum lipids [27,47], white blood cell [30,48] and platelet counts [49,50], liver enzymes [51,52], reduced renal function [29,53,54], cardiovascular diseases [55], type 2 diabetes mellitus [56,57], and male sex [14]. To our knowledge, the observations that serum homocysteine level is higher in the right-sided colorectal tumors and the normalization of serum total protein level entails a constant homocysteine level, were never described previously. Right-sided tumors are known for their increased disease severity [58], therefore, the observation above fits into the knowledge so far; additionally, the latter may be also related to lighter disease severity/lower tumor burden, with high probability.

Our knowledge about the relationship between homocysteine and vitamin D is very limited. In health, the relationship between the two is inversely proportional [14], lower homocysteine level is associated with higher vitamin D_3_ concentrations. Furthermore, a study [59] investigating the supplementation of healthy subjects with a combination therapy of vitamins (500 µg folic acid, 500 µg vitamin B_12_, 50 mg vitamin B_6_, 1200 IU vitamin D, and 456 mg calcium) found that after a 1-year supplementation period, homocysteine level reduced by approximately one-third from the baseline. In other cancerous diseases than CRC, it was found that gastric lymphoma patients requiring total gastrectomy have higher homocysteine and lower vitamin D level than those who only required partial gastrectomy or were not operated on at all [60]. A study of breast cancer patients [61] reported that the homocysteine level of all of the patients was over the median of the normal range (5 to 15 μmol/L) and almost every second patient had a vitamin D deficiency. Another study of breast cancer patients [62] reported significantly lower vitamin D, but the same homocysteine levels, compared to those of healthy subjects. In the case of CRC, it has to be highlighted that we were the first to investigate the relationship between serum homocysteine and vitamin 1,25(OH)_2_D_3_ and their combined effect on CRC. It was found that neither of the two affected each other’s change with the course of the disease, and a positive and negative effect on patient survival was found in a multivariate setting for vitamin 1,25(OH)_2_D_3_ and homocysteine, respectively.

### Limitations

Limitations of the current research were as follows: the retrospective design of the study, the relatively low sample size, and the heterogeneity of CRC population. The former prevented us from properly monitoring the patient compliance of vitamin D supplementation and we could only rely on the available documentation. Furthermore, no intervention was possible if any change in vitamin 1,25(OH)_2_D_3_ or homocysteine levels were indicated. Further limitation related to the design of the study was that the number of patient visits was not equal for all of the study subjects, therefore, to reduce the resulting biases, we chose statistical methods that can robustly address the problem of the missing values.

## 5. Conclusions

Summarizing the results of our retrospective longitudinal observational cohort study, we observed a swoosh-shaped trend for serum homocysteine level changes in all CRC patients, regardless of any clinicopathological features. While the presence of metastases cancelled out all of its positive effects, vitamin D_3_ supplementation had the most benefits in patients without metastases. Although the different directions in the change in serum vitamin 1,25(OH)_2_D_3_ and homocysteine levels appeared to be proportionally opposite to each other, especially in the first interval of our observation, no significant correlation could have been justified between the two parameters. The strong associations found with the comorbidities, serum lipids, and the other laboratory parameters refer to the need for a more complex oncology that is more dependent on interdisciplinary solutions. Limitations of our work are the relatively small sample size and the retrospective design.

Based on literature data [35,36] and our results we propose the following hypothesis. In metastatic CRC, the daily dose of vitamin D_3_ supplementation should be increased until an appropriate titration level is reached, e.g., when it continuously stays within the middle of the normal range. The effectiveness of the treatment should be monitored by the regular measurement of vitamin D_3_ metabolites. Changes to clinicopathological parameters and decreasing vitamin D_3_ levels, despite the persistently high supplementation dosage, may indicate the adjustment of treatment. In order to support and properly assess the role of the hypothesized treatment above, a randomized clinical trial is needed.

## Figures and Tables

**Figure 1 cancers-14-00658-f001:**
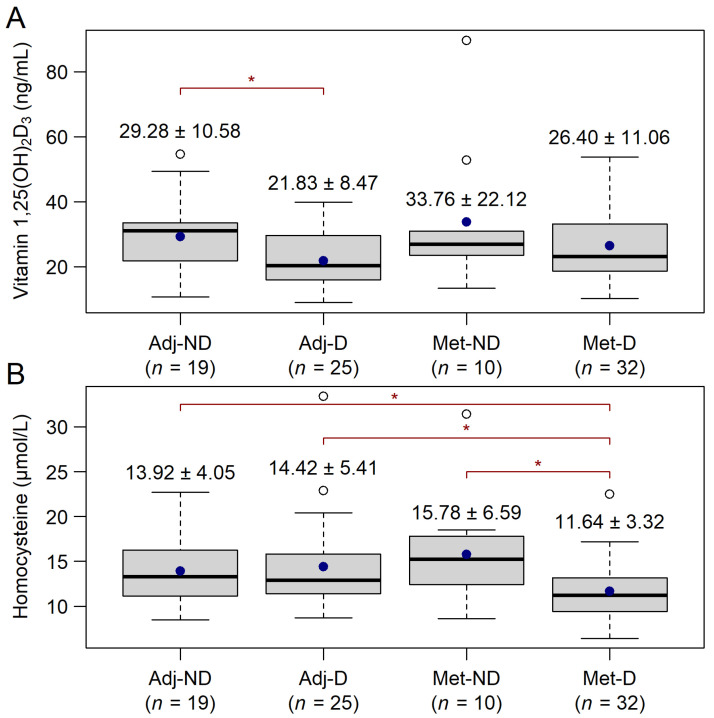
Baseline vitamin 1,25(OH)_2_D_3_ (**A**) and homocysteine (**B**) level of study participants, prior to vitamin D_3_ supplementation (mean ± SD). Without *p*-value correction, vitamin 1,25(OH)_2_D_3_ level differed between the Adj-ND and Adj-D groups (crude *p* = 0.0220, pairwise Wilcoxon–Mann–Whitney U-test), while homocysteine of patients within the Met-D group was significantly lower than those within the other three groups (crude *p*-values: *p* = 0.0400 vs. Adj-ND, *p* = 0.0360 vs. Adj-D and *p* = 0.0470 vs. Met-ND, pairwise Wilcoxon–Mann–Whitney U-test). Adj-ND: patients without metastasis and no vitamin D_3_ supplementation; Adj-D: patients without metastasis with vitamin D_3_ supplementation; Met-ND: patients with metastasis and no vitamin D_3_ supplementation; Met-D: patients with metastasis with vitamin D_3_ supplementation. The hollow black circles, blue dot, and the thick line represent outliers (>1.5 times the interquartile range above the upper quartile), the mean and median value, respectively. * *p* < 0.05.

**Figure 2 cancers-14-00658-f002:**
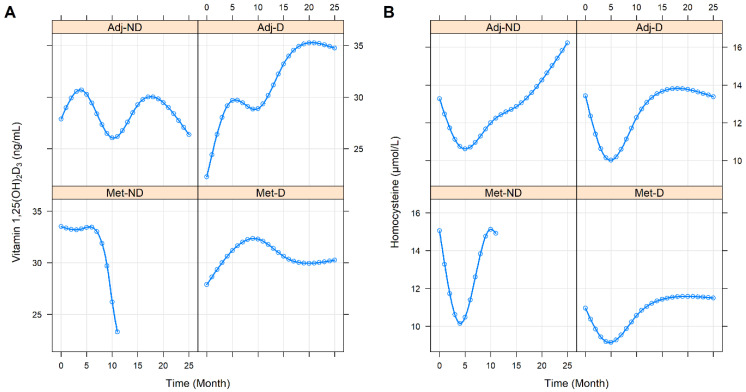
Predictions of average longitudinal changes in vitamin 1,25(OH)_2_D_3_ (**A**) and homocysteine (**B**) level of the study groups throughout the first 25 months of the observation time. Adj-ND: patients without metastasis and no vitamin D_3_ supplementation; Adj-D: patients without metastasis with vitamin D_3_ supplementation; Met-ND: patients with metastasis and no vitamin D_3_ supplementation; Met-D: patients with metastasis with vitamin D_3_ supplementation.

**Figure 3 cancers-14-00658-f003:**
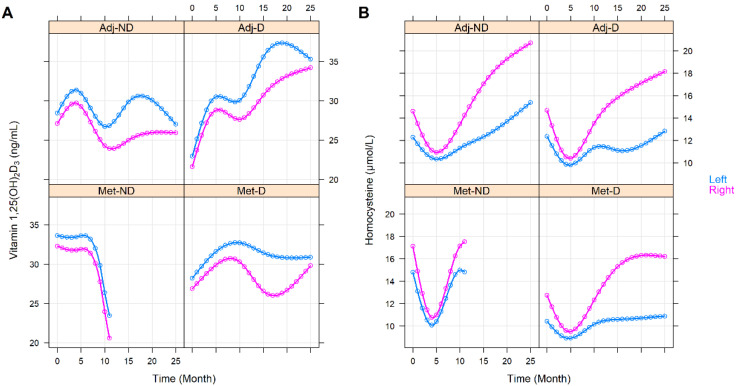
Predictions of average longitudinal changes in vitamin 1,25(OH)_2_D_3_ (**A**) and homocysteine (**B**) level of the study groups throughout the first 25 months of the observation time, stratified by the sidedness of the tumor. Adj-ND: patients without metastasis and no vitamin D_3_ supplementation; Adj-D: patients without metastasis with vitamin D_3_ supplementation; Met-ND: patients with metastasis and no vitamin D_3_ supplementation; Met-D: patients with metastasis with vitamin D_3_ supplementation.

**Figure 4 cancers-14-00658-f004:**
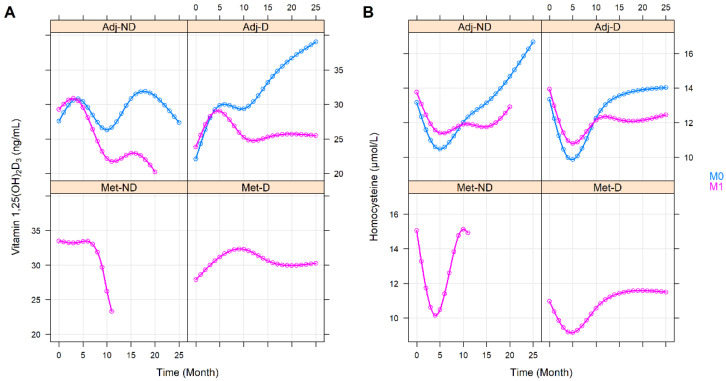
Predictions of average longitudinal changes in vitamin 1,25(OH)_2_D_3_ (**A**) and homocysteine (**B**) level of the study groups throughout the first 25 months of the observation time, stratified by metastases that developed at a later stage of the disease. M0: without metastasis; M1: with metastasis. Adj-ND: patients without metastasis and no vitamin D_3_ supplementation; Adj-D: patients without metastasis with vitamin D_3_ supplementation; Met-ND: patients with metastasis and no vitamin D_3_ supplementation; Met-D: patients with metastasis with vitamin D_3_ supplementation.

**Figure 5 cancers-14-00658-f005:**
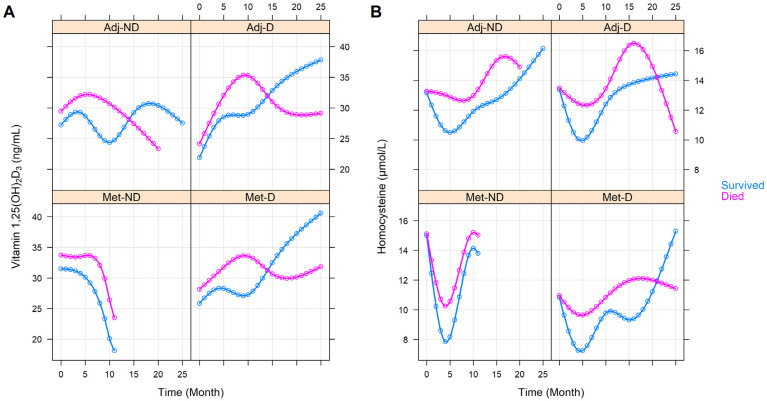
Predictions of average longitudinal changes in vitamin 1,25(OH)_2_D_3_ (**A**) and homocysteine (**B**) level of the study groups throughout the first 25 months of the observation time, stratified by death. Adj-ND: patients without metastasis and no vitamin D_3_ supplementation; Adj-D: patients without metastasis with vitamin D_3_ supplementation; Met-ND: patients with metastasis and no vitamin D_3_ supplementation; Met-D: patients with metastasis with vitamin D_3_ supplementation.

**Figure 6 cancers-14-00658-f006:**
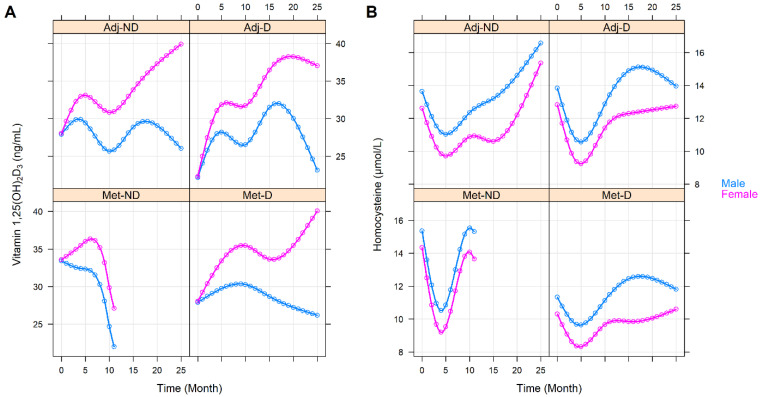
Predictions of average longitudinal changes in vitamin 1,25(OH)_2_D_3_ (**A**) and homocysteine (**B**) level of the study groups throughout the first 25 months of the observation time, stratified by sex. Adj-ND: patients without metastasis and no vitamin D_3_ supplementation; Adj-D: patients without metastasis with vitamin D_3_ supplementation; Met-ND: patients with metastasis and no vitamin D_3_ supplementation; Met-D: patients with metastasis with vitamin D_3_ supplementation.

**Figure 7 cancers-14-00658-f007:**
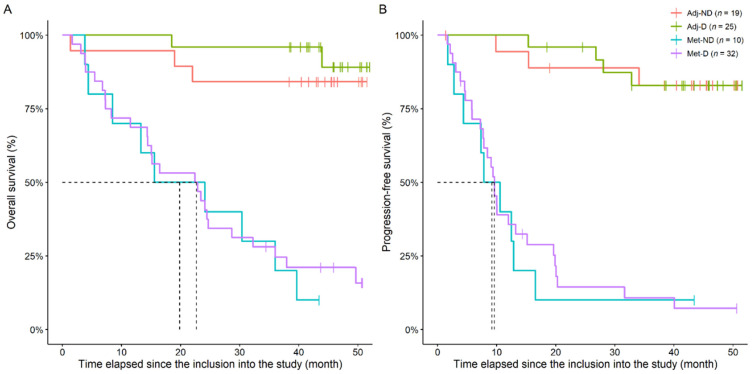
Naïve overall (**A**) and progression-free (**B**) Kaplan–Meier survival curves of the study cohorts. Adj-ND: patients without metastasis and no vitamin D_3_ supplementation; Adj-D: patients without metastasis with vitamin D_3_ supplementation; Met-ND: patients with metastasis and no vitamin D_3_ supplementation; Met-D: patients with metastasis with vitamin D_3_ supplementation. Dashed line represents median survival times.

**Table 1 cancers-14-00658-t001:** Demographic and clinical characteristics of study participants. The significant differences found in staging (*p* < 0.0001), lineage of chemotherapy (*p* < 0.0001), and the occurrence of death/progression (*p* < 0.0001) was related only to the presence of metastases. Unit of frequency data is the number of observations (percentage).

Parameter	Adj-ND (*n* = 19)	Adj-D (*n* = 25)	Met-ND (*n* = 10)	Met-D (*n* = 32)
Age (years)	64.87 ± 10.37	62.58 ± 8.62	65.59 ± 7.16	59.49 ± 11.96
Sex (Male:Female)	12:7 (63.2%:36.8%)	15:10 (60.0%:40.0%)	7:3 (70.0%:30.0%)	20:12 (62.5%:37.5%)
Location of the tumor				
- Right-sided	8 (42.1%)	12 (48.0%)	1 (10.0%)	8 (25.0%)
- Left-sided ^1^	11 (57.9%)	13 (52.0%)	9 (90.0%)	24 (75.0%)
Staging (AJCC [22]) ^2^				
- Stage I	0 (0.0%)	2 (8.0%)	0 (0.0%)	0 (0.0%)
- Stage II	8 (42.1%)	10 (40.0%)	0 (0.0%)	0 (0.0%)
- Stage III	11 (57.9%)	13 (52.0%)	0 (0.0%)	0 (0.0%)
- Stage IV	0 (0.0%)	0 (0.0%)	10 (100.0%)	32 (100.0%)
Distant metastasis developed later with the course of the disease	3 (15.8%)	4 (16.0%)	–	–
Chemotherapy				
- Adjuvant	17 (89.5%)	22 (88.0%)	0 (0.0%)	0 (0.0%)
- Metastatic				
o First line	2 (10.5%)	3 (12.0%)	4 (40.0%)	12 (37.5%)
o Second line	0 (0.0%)	0 (0.0%)	1 (10.0%)	10 (31.25%)
o Third line or above	0 (0.0%)	0 (0.0%)	5 (50.0%)	10 (31.25%)
Medical history				
- Type 2 diabetes mellitus	6 (31.6%)	6 (24.0%)	2 (20.0%)	5 (15.6%)
- Hypertension	12 (63.2%)	16 (64.0%)	9 (90.0%)	18 (56.3%)
- Cardiovascular diseases	3 (15.8%)	6 (24.0%)	3 (30.0%)	4 (12.5%)
- Cardiovascular event(s)	2 (10.5%)	3 (12.0%)	0 (0.0%)	4 (12.5%)
- Thyroid diseases ^3^	2 (10.5%)	3 (12.0%)	3 (30.0%)	4 (12.5%)
Disease progression ^4^	3 (15.8%)	4 (16.0%)	9 (90.0%)	28 (87.5%)
CRC-related death ^5^	1 (5.3%)	1 (4.0%)	9 (90.0%)	25 (78.1%)

Adj-ND: patients without metastasis and no vitamin D_3_ supplementation; Adj-D: patients without metastasis with vitamin D_3_ supplementation; Met-ND: patients with metastasis and no vitamin D_3_ supplementation; Met-D: patients with metastasis with vitamin D_3_ supplementation. ^1^ Three, four, three, and twelve were located in the rectum in each group, respectively. ^2^ Staging given at the time of inclusion to the study. ^3^ Euthyroid status of patients was a required criteria to be included in the study. ^4^ Any progression that occurred between the inclusion date into the study and the latest observation date. ^5^ Deaths related to other diseases than CRC were treated as separate events.

**Table 2 cancers-14-00658-t002:** Baseline laboratory measurements of study participants. Besides a few parameters affected by the presence of metastases, laboratory parameter measurement data suggested that the study population was homogenous at the time of inclusion.

Parameter	Adj-ND (*n* = 19)	Adj-D (*n* = 25)	Met-ND (*n* = 10)	Met-D (*n* = 32)
White blood cell count (10^9^/L)	7.51 ± 3.17	6.80 ± 1.70	8.15 ± 2.45	7.73 ± 3.22
Red blood cell count (10^12^/L)	4.52 ± 0.48	4.55 ± 0.47	4.35 ± 0.37	4.61 ± 0.52
Hemoglobin (g/L)	124.95 ± 18.32	126.04 ± 15.27	127.00 ± 19.09	130.38 ± 18.17
Hematocrit (L/L)	0.38 ± 0.05	0.39 ± 0.04	0.38 ± 0.04	0.39 ± 0.05
Platelet count (10^9^/L)	231.26 ± 60.30	269.96 ± 86.74	332.70 ± 130.46	294.25 ± 132.94
Aspartate transaminase (U/L)	22.11 ± 4.70	24.58 ± 8.60	32.90 ± 19.54	50.00 ± 73.38
Alanine transaminase (U/L)	21.00 ± 9.46	23.12 ± 13.26	27.30 ± 20.47	38.50 ± 32.42
Gamma-glutamyl transferase (U/L)	34.50 ± 17.84	30.42 ± 12.83	155.50 ± 261.67	137.53 ± 163.50
Lactate dehydrogenase (U/L)	182.68 ± 28.79 *	178.50 ± 38.61 *	357.00 ± 291.13 *	587.44 ± 1183.24 *
Creatinine (µmol/L)	79.00 ± 18.95	69.04 ± 14.06	68.80 ± 21.66	64.69 ± 12.67
Estimated glomerular filtration rate $\left(\frac{\mathrm{mL}}{\min\cdot 1.73m^{2}}\right)$	82.74 ± 18.38	90.71 ± 14.03	89.52 ± 21.13	97.48 ± 13.10
Total cholesterol (mmol/L)	5.16 ± 1.24	5.50 ± 1.28	6.35 ± 2.24	5.59 ± 1.45
High-density lipoprotein (mmol/L)	1.54 ±0.51	1.45 ± 0.33	1.37 ± 0.34	1.37 ± 0.44
Low-density lipoprotein (mmol/L)	3.05 ± 0.80	3.39 ± 0.93	3.92 ± 1.43	3.54 ± 1.03
Triglyceride (mmol/L)	1.39 ± 0.99	1.64 ± 0.82	1.63 ± 0.89	1.58 ± 0.52
High-sensitivity C-reactive protein (mg/L)	3.93 ± 3.70 *	3.57 ± 3.50 *	26.90 ± 35.16 *	21.80 ± 43.92 *
Total protein (g/L)	75.58 ± 3.87	76.40 ± 4.55	73.98 ± 3.98	72.98 ± 5.35
Albumin (g/L)	44.24 ± 2.46	44.30 ± 3.13	42.53 ± 4.68	42.94 ± 3.48
Carcinoembryonic antigen (ng/mL)	2.33 ± 1.27 *	2.01 ± 1.48 *	117.21 ± 162.72 *	367.81 ± 1496.30 *
Carbohydrate antigen 19-9 (U/mL)	6.53 ± 11.39 *	7.46 ± 12.98 *	451.60 ± 813.46 *	925.56 ± 3656.01 *

Adj-ND: patients without metastasis and no vitamin D_3_ supplementation; Adj-D: patients without metastasis with vitamin D_3_ supplementation; Met-ND: patients with metastasis and no vitamin D_3_ supplementation; Met-D: patients with metastasis with vitamin D_3_ supplementation. * Differences in high-sensitivity C-reactive protein (Kruskal–Wallis test, *p* = 0.0025), lactate dehydrogenase (*p* = 0.0004), carcinoembryonic antigen (*p* < 0.0001), and in carbohydrate antigen 19-9 (*p* = 0.0080) were found to be significant only between patients with and those without metastasis, while all measured parameters were the same between the non-supplemented and vitamin D supplemented groups.

## Data Availability

The datasets used and/or analyzed during the current study are available from the corresponding author on reasonable request.

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
