# Peer review of "Longitudinal Analysis of 1α,25-dihidroxyvitamin D_3_ and Homocysteine Changes in Colorectal Cancer"

_cancers, 2022, doi:10.3390/cancers14030658_

Round 1

Reviewer 1 Report

In this manuscript Mühl et al. analyzes the effect of vitD3 supplementation on early stage and metastasized CRC patients –measuring vitD3 and homocysteine levels in these patients.  From the study VitD deficiency seems to increase homocysteine levels in non-metastasized early stage CRC patients; but not in metastasized cancer. However, survival is not affected by vitD3 supplementation. The authors also explore homocysteine levels in these patients classifying them based on sidedness of the tumor and on presence of comorbidities like type II diabetes, hypertension etc. It is overall a comprehensive analysis of patient data though the sample size is small

Some specific questions:

 Some Does serum vit D levels affect homocysteine levels in control samples (individuals without CRC)? Please add some comments on correlation of vitD3 and homocysteine levels in other cancersif there are any reports

Does vitD3 supplementation affect tumor size?

The parameters which are significantly correlated by vitD supplementation (eg: cardiovascular disease, sex) can be moved to the main figures and the  parameters which do not show any involvement with vitD3/homocysteine can be moved to the supplementary material.

Figure 1: It will be better if you indicate within the graphs (with *) which differences are statistically significant and indicate in the legend the p values and type of statistical tests used.

Table 2: It will be easier for the reader if you highlight the parameters that changed between patients with metastasis and those without it. Since vitD does not seem to affect any of these parameters you may consider moving this table to supplementary data

Reviewer 2 Report

The article by Mühl et. al. entitled “Longitudinal Analysis of 1α,25-dihidroxyvitamin D3 and Homocysteine Changes in Colorectal Cancer” describes a retrospective study aimed at understanding the influence of vitamin D3 and homocysteine circulating levels on cancer progression and survival in CRC patients.  The results of the study appear to align with previous work in the field and uncovers an interesting relation between homocysteine and CRC sidedness that has not been described previously.  We provide several comments below that we hope will help the authors in their revision of the manuscript.  Our major concern is the rational for the vitamin D3 supplementation provided by the authors (comment #8).

  1. Line 49: A recent report on Vitamin D3 influence on Th1 cells could potentially be added https://doi.org/10.1038/s41590-021-01080-3
  2. Line 54: “most” should be “largest”
  3. Line 67: “no study investigated neither the single-time or …” could be better phrased as “no study investigated the single-time nor …”
  4. Line 123: What is meant by “fact”?
  5. Line 127: “sperate” should be “separate”
  6. Lines 342-368 (Discussion) and Introduction: It is a bit ambiguous what the prior clinical and investigational work on vitamin D3 supplementation has revealed. The authors mention correlations between vitamin D3 serum levels and overall patient survival with CRC, but the influence of vitamin D3 supplementation is less clear.  Can the authors provide some values on what prior clinical and/or investigational work has found to be sufficient vitamin D3 supplementation and serum levels to improve CRC survival?  It would greatly aid in putting the work the authors report here in context with previous work.
  7. Lines 394-395: “CRC is a known” should be “CRC is known”
  8. Lines 468-471: The authors provide recommendations for increasing vitamin D3 supplementation until the “appropriate titration level is reached”. The authors provide further evidence for this recommendation either based on the study described or previous literature?  Also can the authors comment on what the “appropriate titration level” is and how a clinician would know it has been reached?
